# Therapeutic Cancer Vaccination with Immunopeptidomics-Discovered Antigens Confers Protective Antitumor Efficacy

**DOI:** 10.3390/cancers13143408

**Published:** 2021-07-07

**Authors:** Karita Peltonen, Sara Feola, Husen M. Umer, Jacopo Chiaro, Georgios Mermelekas, Erkko Ylösmäki, Sari Pesonen, Rui M. M. Branca, Janne Lehtiö, Vincenzo Cerullo

**Affiliations:** 1Drug Research Program, Division of Pharmaceutical Biosciences, Faculty of Pharmacy, University of Helsinki, 00790 Helsinki, Finland; karita.peltonen@helsinki.fi (K.P.); sara.feola@helsinki.fi (S.F.); jacopo.chiaro@helsinki.fi (J.C.); erkko.ylosmaki@helsinki.fi (E.Y.); 2Helsinki Institute of Life Science (HiLIFE), University of Helsinki, 00790 Helsinki, Finland; 3Translational Immunology Research Program (TRIMM), University of Helsinki, 00790 Helsinki, Finland; 4iCAN Digital Precision Cancer Medicine Flagship, University of Helsinki, 00790 Helsinki, Finland; 5Science for Life Laboratory, Department of Oncology-Pathology, Karolinska Institutet, 171 65 Solna, Sweden; husen.umer@ki.se (H.M.U.); georgios.mermelekas@ki.se (G.M.); rui.mamede-branca@ki.se (R.M.M.B.); 6Valo Therapeutics Oy, 00790 Helsinki, Finland; sari.pesonen@valotx.com

**Keywords:** tumor antigen, endogenous retrovirus, cancer vaccine, immunotherapy, immunopeptidome, ligandome, breast cancer, mass spectrometry, oncolytic vaccine

## Abstract

**Simple Summary:**

Immunotherapy has revolutionized cancer treatment, yet many tumors remain resistant to current immuno-oncology therapies. Here we explore a novel, customized oncolytic adenovirus vaccine platform as immunotherapy in a resistant tumor model. We present a workflow for customizing the oncolytic vaccine for improved tumor targeting. This targeting is based on experimentally discovered tumor antigens, which are incorporated as active components of the vaccine formulation. The pipeline may be further applied for designing personalized therapeutic cancer vaccines.

**Abstract:**

Knowledge of clinically targetable tumor antigens is becoming vital for broader design and utility of therapeutic cancer vaccines. This information is obtained reliably by directly interrogating the MHC-I presented peptide ligands, the immunopeptidome, with state-of-the-art mass spectrometry. Our manuscript describes direct identification of novel tumor antigens for an aggressive triple-negative breast cancer model. Immunopeptidome profiling revealed 2481 unique antigens, among them a novel ERV antigen originating from an endogenous retrovirus element. The clinical benefit and tumor control potential of the identified tumor antigens and ERV antigen were studied in a preclinical model using two vaccine platforms and therapeutic settings. Prominent control of established tumors was achieved using an oncolytic adenovirus platform designed for flexible and specific tumor targeting, namely PeptiCRAd. Our study presents a pipeline integrating immunopeptidome analysis-driven antigen discovery with a therapeutic cancer vaccine platform for improved personalized oncolytic immunotherapy.

## 1. Introduction

Immunotherapy has reshaped the treatment of highly immunogenic tumors as exemplified by the success of immune checkpoint blockade for metastatic melanoma. Recent evidence implies triple-negative breast cancer (TNBC) may also be susceptible to immunotherapy: it shows high mutational load among breast cancers, expression of checkpoint molecules such as PD-L1, and infiltration of immune cells suggestive of pre-existing immunity towards tumor. A gene expression profiling study of over 3000 cases further described an immunomodulatory subtype of TNBC enriched for genes involved in immune-related cellular processes [1]. Importantly, these immunological biomarkers associate with better clinical outcome of the patients [2,3]. In a recent phase 3 clinical trial, checkpoint inhibition in combination with paclitaxel prolonged progression-free survival of metastatic TNBC patients [4], leading to the FDA approval of the combination treatment for PD-L1-positive tumors. Despite these outstanding achievements, only a relatively select group of TNBC patients may fully benefit from the treatment. The high response rate to immunotherapy correlates with high tumor PD-L1 expression, which is present in 19% of TNBC specimens [5]. Furthermore, the tumor infiltrating lymphocytes may not target a clinically effective tumor antigen, limiting the feasibility of immune checkpoint blockade as single therapy. Combining checkpoint inhibition with targeted activation of tumor antigen-specific T-cells, as achieved with therapeutic cancer vaccination, may have the potential to broaden the applicability of checkpoint blockade therapy beyond the currently targeted patient group.

The cornerstone of effective immunotherapy is the recognition of tumor antigens by cytotoxic T-lymphocytes. On estimate, thousands of tumor antigens, short peptides displayed in the context of MHC class I molecules (MHC-I) on the tumor cell surface, can be naturally presented [6]. Knowledge of the presented tumor antigen repertoire, different from that of normal cells, is necessary for successful design of more targeted treatment modalities [7]. For several tumor types, including TNBC, information on well established, clinically relevant tumor antigens is currently limited. For tumor types with lower mutational burden, tumor-associated antigens may offer an attractive alternative to broaden the repertoire of clinically feasible targets [7,8]. These tumor-associated antigens are derived from either tissue-specific or overexpressed proteins in tumor, or proteins expressed in tumor due to epigenetic changes (e.g., cancer testis antigens). Furthermore, endogenous retroviruses (ERVs) are emerging as an interesting source of tumor antigens as their expression in tumor positively associates with immune cell infiltration and even with immune therapy response [9,10,11].

Here, we identify tumor antigens and investigate their applicability as personalized cancer vaccination in a preclinical model of TNBC. We identify the tumor-associated antigen repertoire, tumor immunopeptidome, combining MHC class-I immunoaffinity purification with state-of-the-art mass spectrometry analysis. Among the MHC-I presented immunopeptidome we discovered a novel peptide originating from an endogenous retrovirus protein. We further explored the potential of targeting selected tumor antigens using prophylactic or therapeutic vaccination in an experimental mouse model of TNBC. Our results support the notion that tumors with lower mutational burden can also be susceptible to targeted immunotherapy. Furthermore, we present a pipeline for improved cancer immunotherapy based on mass spectrometry discovery of therapeutically targetable T-cell epitopes combined with oncolytic cancer vaccines (Figure 1a).

## 2. Materials and Methods

### 2.1. Cell Lines

As the mouse model of TNBC, we chose the 4T1 mouse mammary tumor cell line, a highly metastatic TNBC type originally derived from a spontaneously arising tumor in Balb/c mice. The cell line was purchased from ATCC (Manassas, VA, USA) and cultured according to ATCC recommendations in DMEM supplemented with 10% heat-inactivated FBS and antibiotics in +37 °C, 5% CO_2_ in a humidified atmosphere.

### 2.2. Purification of MHC Class-I Complexes

MHC-I peptide complexes were immunoaffinity purified from the 4T1 mouse tumor cell line using MHC-I antibody against H-2Kd/H-2Dd (clone 34-1-2S, SouthernBiotech, Birmingham, AL, USA) and applying the method by Bassani-Sternberg [12] with minor modifications. Frozen 4T1 cells (1–3 × 10^8^) were lysed with 0.25% sodium deoxycholate, 0.2 mM iodoacetamide, 1 mM EDTA, 1 mM PMSF, and 1% octyl-β-D glucopyranoside in the presence of protease inhibitors in PBS at 4 °C for 2 h. The lysate was precleared (2000× *g*, 5 min at 4 °C) and cleared by centrifugation at 20,000× *g*, 40 min at 4 °C prior to loading to the immunoaffinity column (AminoLink, Pierce) with covalently linked antibody. Following binding, the affinity column was washed using 7 column volumes of each buffer (150 mM NaCl, 20 mM Tris-HCl; 400 mM NaCl, 20 mM Tris-HCl; 150 mM NaCl, 20 mM Tris-HCl and 20 mM TrisHCl, pH 8.0), and bound complexes were eluted in 0.1 N acetic acid.

### 2.3. Purification and Concentration of MHC-I Peptides

Eluted peptide MHC-I complexes were desalted using Sep-Pak-C18 cartridges (Waters). Cartridges were prewashed with 80% acetonitrile in 0.1% trifluoro acetic acid (TFA) and with 0.1% TFA prior to loading of the sample and then with 0.1% TFA. The peptides were purified from MHC-I complex by elution with 40% acetonitrile in 0.1% TFA prior to drying the samples using vacuum centrifugation (Eppendorf, Hamburg, Germany).

### 2.4. LC-MS Analysis of MHC-I Peptides

Each dry sample was dissolved in 11 µL of LCMS solvent A (97% water, 3% acetonitrile, 0.1% formic acid), of which 10 µL was injected into a C18 guard desalting column (Acclaim pepmap 100, 75 µm × 2 cm, nanoViper, Thermo, Waltham, MA, USA). After 5 min of flow at 5 µL/min with the loading pump, the 10-port valve was switched to analysis mode, in which the NC pump provided a flow of 250 nL/min through the guard column. The linear gradient then proceeded from 2% solvent B (95% acetonitrile, 5% water, 0.1% formic acid) to 25% B in 90 min followed by wash at 99%B and re-equilibration. Total LC-MS run time was 123 min. We used a nano EASY-Spray column (pepmap RSLC, C18, 2 µm bead size, 100 Å, 75 µm internal diameter, 50 cm long, Thermo) on the nano electrospray ionization (NSI) EASY-Spray source (Thermo) at 60 °C. One raw file (20190121) was generated in the same system but using a shorter column and shorter gradient, 15 cm and 30 min, respectively. Online LC-MS was performed using a hybrid Orbitrap Fusion mass spectrometer (Thermo Scientific, Waltham, MA, USA). FTMS master scans in profile mode with 120,000 resolution (and mass range 300–750 m/z) were followed by data-dependent MS/MS in centroid mode on the top 10 ions using collision-induced dissociation (CID) at 32% normalized collision energy, activation time of 10 ms, and activation Q of 0.25. Precursors were isolated with a 2 m/z window. Automatic gain control (AGC) targets were 1e6 for MS1 and 1e4 for MS2. Maximum injection times were 100 ms for MS1 and 100 ms for MS2. Dynamic exclusion was used with 30 s duration. Only precursors with charge state 2–4 were selected for MS2.

### 2.5. Proteomics Database Search

MS/MS spectra were searched by Byonic v3.6.0 (Protein Metrics Inc., Cupertino, CA, USA), using a target-decoy strategy. For the first search for tumor-associated antigens, the database used was the Uniprot mouse reference protein database (53,378 protein entries, including Swissprot and some TrEMBL entries, downloaded from uniprot.org on 20180917). Precursor mass tolerance of 10 ppm and product mass tolerance of 0.36 Da for CID-ITMS2 were used. No enzyme specificity was used, and oxidation of methionine (common2) and phosphorylation on serine, threonine, or tyrosine (rare2) were used as variable modifications. Maximum precursor mass was 1500, with only 1 precursor per MS2 spectrum allowed, and a smoothing width of 0.01 m/z. False discovery rate (FDR) cutoff of 5% was employed at peptide level.

For the discovery of non-canonical peptides (ERV antigens), we performed re-search against the wider Uniprot database containing all Swissprot and all TrEMBL murine protein entries (restricted to taxonomy 10090, *Mus musculus*, containing a total of 92,607 sequences, downloaded from uniprot.org, accessed on 6 October 2020). The settings were the same as above, except that phosphorylations were not considered as modifications.

We validated the identification of selected endogenous peptides with synthetic peptides and produced mirror plots showing endogenous and respective synthetic peptide MS2 spectra. For each endogenous/synthetic MS2 spectrum pair, quality scores pertaining to the peptide-spectrum match are displayed (Precursor error, DeNovoScore, MSGFScore, SpecEvalue, Evalue). These were obtained by using an MSGF+ [13] search against a database containing only the peptides of interest.

The mass spectrometry proteomics data have been deposited to the ProteomeXchange Consortium via the PRIDE partner repository with the dataset identifier PXD016112.

### 2.6. In Silico Analysis of the MHC-I Peptides

Peptide motif analysis was performed using Gibbs clustering analysis [14] at the DTU bioinformatics server (tool version 2.0). Unsupervised alignment and clustering of input peptides (9 mers, 10 mers or 11 mers separately) as 1–5 clusters and respective motif lengths was carried out using default settings. The known H-2Kd and H-Dd motifs were obtained from DTU Bioinformatics NetMHC 4.0 Motif Viewer [15]. MHC-I binding affinity predictions of 9 mers was performed using the IEDB resource tools (‘IEDB recommended 2.19′ method). To identify MHC-I peptide-enriched biological processes, 9 mers were mapped back to their source proteins, and overrepresentation analysis was performed using PANTHER (version released 20190711) [16]. Mouse proteome was used as background (reference list *Mus musculus*, all genes in database) and ‘GO biological process complete’, ‘GO molecular function complete’, and ‘GO cellular component complete’ (release 3 July 2019) as the annotation data set using Fisher’s exact test with FDR correction.

### 2.7. Mice and Animal Experiment

The animal experiment was approved by the Experimental Animal Committee of Southern Finland National Animal Experiment Board, ELLA, (license number ESAVI/9817/04.10.07/2016) and was performed under the guidelines of the of the Regional State Administrative Agencies in Finland. Age-matched 4–6 weeks old female Balb/c OlaHsd mice (Envigo, Venray, The Netherlands) were used as the syngeneic mouse tumor model of 4T1 TNBC. Mice were immunized three times (15, 12, and 4 days prior to tumor engraftment) by s.c injection of peptide pools plus adjuvant (5–6 peptides per pool, á 25 µg peptide plus 100–125 µg poly I:C (Vaccigrade, InvivoGen, Toulouse, France)). The used peptide pools are as depicted in Table 1. Peptides were purchased as custom synthesis from Zhejiang Ontores Biotechnologies Co. (Hangzhou, Zhejiang, China). To engraft the tumors, 3 × 10^5^ 4T1 cells in PBS were injected s.c. in the right flank of the animals. Tumor growth was measured every second day using a digital caliper once the tumors became palpable, and tumor volume was calculated using the formula: (length × width^2^)/2. Tumors were allowed to grow for 21 days until the mice were sacrificed, and the tumors and spleens were collected for further analysis. Percent tumor growth inhibition (%TGI) was defined as the difference between the median tumor volume (MTV) of the adjuvant and immunization groups, using the formula: %TGI = ((MTVadjuvant-MTVimmunised/MTVadjuvant)) × 100. Additionally, individual mice were scored as responders (at least 30% decrease in tumor volume in comparison to the median tumor volume of the adjuvant group) or non-responder (at least 20% increase in tumor volume in comparison to the median tumor volume of the adjuvant group) at the study endpoint.

For the therapeutic experiment, the PeptiCRAd vaccine platform combining an oncolytic adenovirus and polyK peptide epitope was used in the context of established 4T1 tumors. VALO-mD901 adenovirus ([17] Ad5/3, delta 24, delta E3-CR1-alpha, gp19K and 14.7 K genes) expressing murine *Cd40L* and *Ox40L* under CMV-promoter was utilized as the PeptiCRAd platform. PeptiCRAd formulation was prepared as previously described by complexing the oncolytic adenovirus with polyK peptides for 15 min at room temperature [18] and used within 30 min following complex formation. For TAA-PeptiCRAd, VALO-mD901 was complexed with KKKKKKKFYLETQQQI, KKKKKKSYHPALNAI, and KKKKKKYQAVTATL peptides and for ERV-PeptiCRAd with KKKKKFYLPTIRAV and KKKKKKKTYVAGDTQV peptides. These peptides respond to the initially discovered peptides FYLETQQQI, SYHPALNAI, and KYQAVTATL originating from Brap, Birc6, and Rpl13a source proteins, respectively (experiment shown in Section 3.4) and peptides FYLPTIRAV and TYVAGDTQV originating from the ERV source genes (experiment shown in Section 3.5). Viral dose was 1 × 10^9^ vp complexed with 8 μg peptide (total peptide amount) per tumor. PeptiCRAd was administered intratumorally, anti-PD-1 intraperitoneally, and PBS was used as mock injections. PolyK-tailed peptides were purchased as custom synthesis from Ontores Zhejiang Ontores Biotechnologies Co. (Hangzhou, Zhejiang, China) (tumor-associated antigens) and GeneScript (endogenous retrovirus antigens).

### 2.8. Immunogenicity Analysis

Splenocytes from tumor-bearing animals were harvested at the study endpoint from the therapeutic experiment. Splenocytes were smashed and processed into single cells through a 70 micron cell strainer. Immune cell reactivity towards individual peptides was tested as IFN-gamma secretion upon re-stimulation using ELISPOT (ImmunoSpot, CTL Europe GmbH, Bonn, Germany). A total of 1 × 10^6^ splenocytes were re-stimulated for 2 days with 1 ng individual peptides. ELISPOT plate manufacturer instructions were used for staining of the plate, and the spots were read and quantified by ImmunoSpot (CTL Europe GmbH, Bonn, Germany).

### 2.9. Flow Cytometric Analysis of Tumor-Infiltrating Lymphocytes

The tumors were digested with collagenase and DNAse (1 h, +37 °C) and pushed through a 70 micron cell strainer. Non-specific binding of immunoglobulin to the Fc receptors was blocked with anti-mouse CD16/32 (TruStain Fc block, Biolegend, CA, USA). The tumors were stained using antibodies against: CD3 (clone 145-2C11), CD8 (clone 53-6.7), PD-1 (clone 29F.1A12), and Tim3 (clone B8.2C12) (Biolegend) at +4 °C for 45 min. Flow cytometric acquisition of 1 × 10^6^ events was performed using BD LSRFortessa (BD Biosciences) and was analyzed using FlowJo 10.4 software (Ashland, Willington, DE, USA).

### 2.10. Statistical Analysis

Statistical analysis was performed using GraphPad Prism 6.0 software (GraphPad version 8.2.0 Software, San Diego, CA, USA). For tumor growth curve analysis, 2-way ANOVA with Tukey’s multiple comparison was used, and *p*-values < 0.05 were considered as statistically significant. For flow cytometric data analysis, an unpaired t test was used. Results are expressed as mean of the group ± standard error mean (SEM).

## 3. Results

### 3.1. Direct Identification of Tumor-Associated Antigens in Mouse Triple-Negative Breast Tumor

Therapeutic cancer vaccination using neoantigens has demonstrated clinical benefit in highly immunogenic tumors with high levels of somatic mutations such as melanoma. In tumor types with low mutational burden, the likelihood of identifying neoepitopes amenable for therapeutic targeting is more limited. This limitation calls for identifying novel, effective, shared tumor-associated antigens for tumor types susceptible for immunotherapy in order to expand the current immunomodulatory strategies to benefit wider patient groups. In this regard, TNBC is an attractive model with shown sensitivity to checkpoint blockade.

To identify naturally presented tumor-associated antigens for a TNBC model, we performed immunopeptidome analysis on MHC class I (MHC-I) bound peptides using the murine 4T1 cell line (Appendix A). MHC-I peptide complexes were immunoaffinity purified, and eluted peptides were analyzed by tandem mass spectrometry. Using the murine reference proteome (uniprot.org 53,378 entries) as the search database, we found a total of 2207 MHC-I-associated peptide sequences using an FDR threshold of 5% for peptide identification. The depth of our immunopeptidome analysis is in line with recent publications identifying H-2K^b^ and H-2D^b^ restricted immunopeptidomes for several murine tumor cell lines and mice tissues [19,20]. Next, we performed bioinformatics analysis of the identified MHC-I peptides and their source proteins (Appendix A). The peptides represent the typical length distribution of isolated MHC-I ligandomes (Figure 1b) with 9 mers as the most enriched peptide species. Of the unique 9 mers, 75% are known H-2Kd or H-2Dd ligands (described as *Mus musculus* epitopes in IEDB database 20210512). Gibbs clustering of the 9 mers, 10 mers, and 11 mers showed the peptides clustered into two distinct groups with preference for reduced amino acid complexity for residues at positions P2 and Ω or P2, P3, and Ω (Figure 1c and Appendix A). These data are in line with known H-2K^d^ and H-2D^d^ binding motifs with respect to their anchor residue positions and amino acid characteristics. Due to the high sensitivity of the mass spectrometry, we identified also longer peptides (12 mers–14 mers), which may additionally represent contaminating co-purifying peptides among a few ‘true’ MHC-I ligands.

We next aimed to identify the high-confidence MHC-I ligands among the 9 mers. Prediction of the peptide binding affinity to MHC-I showed 67% of the 9 mers bound either H-2 Kd or H-2 Dd and can thus be considered as MHC-I ligands (using the <500 nM IC50 affinity value as cutoff for MHC-I binding, Figure 1d). The vast majority of these ligands showed clear preference to H-2Kd, and 41% of the H-2Kd ligands bound H-2Kd with high affinity (<50 nM IC50, Appendix A).

The MHC-I-presented peptides are derived from the degradation of cellular proteins, and as such the immunopeptidome reflects the cellular events or cell status. During malignant transformation, various cellular processes become deregulated [21,22] leading to abnormal protein production. The presented peptide ligands derived from deregulated proteins may offer an attractive target for immunotherapy, as they may present tumor-selective epitopes altered, e.g., in abundance in comparison to normal cells. In our 4T1 immunopeptidome, the identified 9 mer ligands are derived from various different source proteins with the majority of source proteins producing only one detectable MHC-I ligand. We observed few exceptions, which showed a wider presentation coverage (maximum of 4 assigned MHC-I ligands). The source proteins are enriched in nuclear and intracellular proteins with various enzymatic or nucleotide binding activities. The source proteins have a function in various biological processes, mainly reflecting the high proliferative, metabolic, and biosynthetic status of the cells (Figure 1e).

### 3.2. Discovery of MHC-I-Presented Endogenous Retroviral Antigen

A particularly interesting class of tumor antigens may arise from aberrant translation in tumor tissue. These tumor-selective protein products are generated from genomic loci previously considered as non-coding, intronic, or are products of un-annotated alternative gene translations. Remarkably, the presence of aberrant translation products has recently been demonstrated in various tumor types, including human breast tumors [23]. Importantly, these candidate tumor antigens can generate specific T-cell responses towards the tumor with clinical relevance [11,24].

We thus re-searched our mass spectrometry ligandome data files against a database containing the entire unreviewed TrEMBL set from Uniprot, a total of 92,607 protein entries (all *Mus musculus* Swissprot + TrEMBL entries) (Appendix A). Many of the TrEMBL entries that do not map to canonical genes are potentially the product of aberrant transcription and translation in tumors. Applying an FDR cutoff of 5%, as before, resulted in 2481 MHC-peptides found (Appendix A). These peptides showed typical MHC-I ligand characteristics in terms of peptide length, amino acid enrichment, and predicted binding efficiency to MHC-I. However, this allowed the discovery of additional MHC peptides mapping to TrEMBL entries (including peptides not found in the reference proteome searches), most of which actually correspond to canonical proteins, but one belonged to theoretically non-coding genomic regions. This peptide, FYLPTIRAV (confidence score |log prob| = 4.75) from hypothetical viral gag protein Q811J2 (Uniprot accession), maps to an endogenous retroviral (ERV) region, LOC72520. We further validated the identification of FYLPTIRAV peptide with a synthetic one, and the mirror plots showing endogenous peptide and respective synthetic peptide are presented as Appendix A.

### 3.3. Control of Tumor Growth by MHC-I Ligand Immunization

Tumor regression can be induced by therapeutic cancer vaccination targeting tumor-selective epitopes, and in experimental tumor models marked responses are achievable even when using a single immunodominant epitope. In case of a highly heterogeneous tumor, such as TNBC, the identification of immunodominant epitopes able to target the majority of the tumorigenic cells may represent a challenge. Thus, tumor vaccines can be designed as multivalent to obtain protective T-cell responses against a larger repertoire of tumor antigens for eliciting sustainable immunotherapy responses.

We studied the antitumorigenic potential of peptide-based vaccination utilizing the identified tumor associated antigens in BALB/c mice. The tumor antigens for the vaccine formulations were selected according to predicted high affinity of peptide binding to MHC-I. Additionally, peptides originating from source proteins having a function in cancer biology (such as Ski) were included despite showing lower putative MHC-I binding (Table 1). Additionally, one group (peptide pool #4) was selected based on an in-house developed tool to estimate peptide immunogenicity using peptide similarity to viral epitopes [25]. In short, peptide similarity to viral peptides was computed via pairwise weighted alignment. Peptide central section similarity was prioritized, as being mostly involved in peptide/TCR interaction. Moreover, peptide putative binding affinity to MHC-I was included within the immunogenicity estimate. Hence, the peptide pool 4 peptides were chosen based on similarity to viral peptides and high putative binding to MHC-I as surrogate to peptide immunogenicity.

Mice were immunized thrice with peptide pools (5–6 MHC-I restricted peptides per pool, Table 1) using poly I:C as an adjuvant. An irrelevant peptide plus adjuvant was used as mock control.

Following immunization, mice were challenged with 4T1 tumor cells, and tumor growth was measured (Figure 2a). We observed reduced tumor growth in two immunization groups (Peptide pool 3 and Peptide pool 4) with 44% and 36% tumor growth inhibition at the endpoint in comparison to the adjuvant control (Figure 2b). The effect in tumor growth was statistically significant in these immunization groups, with 62.5% and 50% of individual mice showing antitumorigenic responses, respectively (Figure 2c and Appendix A). In Peptide pool 1 two mice developed small tumors following immunization. However, as a group, the effect on tumor control was not markedly different from the adjuvant controls. Of note, one peptide immunization group (Peptide pool2) showed larger tumors than the adjuvant group. This undesirable effect emphasizes the need to acquire sufficient knowledge on the antigens and their impact on the tumor and immune biology.

The moderate antitumor responses prompted us to study the correlation of the individual responses with frequency and phenotype of tumor-infiltrating immune cells. FACS analysis on tumor infiltrating lymphocyte (TIL) relative frequency showed comparable CD8+ T-cell infiltration among the different treatment groups (on average 0.19–0.34% of the tumor cells stained CD8-positive) (Figure 3a). Peptide vaccination did not increase CD8+ T-cell infiltration in comparison to the adjuvant group, and no clear correlation between individual tumor size and CD8+ cell infiltration was observed (Appendix A). Thus, even though immunization using tumor antigens elicited antitumor control, this was not associated with strong overall increase in immune cell infiltration within the tumors.

Tumor-infiltrating immune cells express multiple co-stimulatory and inhibitory receptors in response to T-cell receptor (TCR) stimulation. Antigen exposure triggers the initial clonal expansion of antigen-specific T-cell populations. However, under persistent exposure to the antigen co-expression of several inhibitory receptors, T-cells are rendered dysfunctional. This T-cell ‘exhaustion’ is observed during chronic viral infection [26,27] as well as in response to tumor antigen exposure, either spontaneous or tumor vaccine-induced [28,29]. Concomitantly, expression of the inhibitory receptors can be used as a surrogate marker for detecting and identifying the tumor reactive repertoire of CD8+ cells from the bulk TILs [30,31]. Especially interesting in this respect is the co-expression of PD-1, LAG-3, and TIM-3 on CD8+ TILs, as these are shown to phenotypically identify tumor-reactive CD8+ lymphocytes, regardless of antigen specificity [30]. We initially analyzed the PD-1-expressing CD8+ cells among the TILs. The vast majority of CD8+ cells expressed PD-1 with no clear increase in the frequency of PD-1+ cells in the vaccination groups versus control (Appendix A). However, the CD8+ cells in the peptide vaccinated mice expressed PD-1 at significantly higher levels in comparison to the adjuvant controls (Figure 3b), suggesting a qualitative difference in these tumor infiltrating cells. A subpopulation of the PD-1+ CD8+ cells additionally co-expressed TIM-3, defining exhausted CD8+ T-cells (Figure 3c). Interestingly, we observed higher prevalence of the PD-1+ Tim-3+ double-positive CD8+ cells associated with smaller tumor size, suggesting this subpopulation of immune cells may be important for tumor control activity (Figure 3d).

### 3.4. Therapeutic Vaccination with PeptiCRAd Cancer Vaccine Targeted to Tumor-Associated Antigens Controls Tumor Growth

We reasoned that the immunosuppressive nature of the tumor model may require direct modulation of the tumor microenvironment, as achieved with oncolytic viruses or checkpoint inhibition, for enhanced T-cell mediated immunity to control tumors [32,33]. We have previously shown increased presence of antigen-specific TILs within tumors with less exhausted phenotype following immunization with a novel cancer vaccine platform PeptiCRAd [17,33]. PeptiCRAd technology combines an oncolytic adenovirus and tumor antigen for enhanced tumor-specific CD8+ T-cell responses towards tumors [18]. Tumor antigen targeting is achieved by adsorbing the tumor peptide onto the oncolytic viral capsid to guide the specificity of the vaccine-induced immune response. Our preliminary results support generation of improved antigen-specific responses with the PeptiCRAd platform over peptide-Poly I:C vaccination (Appendix A).

To further investigate the therapeutic potential of tumor-associated antigen cancer vaccination as immunotherapy, we applied the PeptiCRAd technology. Additionally, we tested the platform in a therapeutically more challenging setting with pre-established tumors. Tumor-bearing mice were vaccinated intratumorally with PeptiCRAd complexed with the identified tumor-associated antigens (TAA-PeptiCRAd group) or with the oncolytic virus (Virus group) alone or in combination with checkpoint inhibition (Virus + a-PD-1 and TAA-PeptiCRAd + a-PD-1) (Figure 4a). The immunopeptidomics discovered peptides FYLETQQQI, SYHPALNAI, and KYQAVTATL originating from Brap, Birc6, and Rpl13a source proteins, respectively, were used as antigens for the TAA-PeptiCRAd. This therapeutic vaccination led to significant protection in TAA-PeptiCRAd-receiving mice over the virus control (Figure 4b,c, *p* < 0.01) with 90% of mice showing potent antitumor activity (Appendix A). The combination of PeptiCRAd with checkpoint inhibition (anti-PD-1 antibody) did not markedly increase the overall therapy response over PeptiCRAd treatment despite conferring the highest systemic T-cell responses against the tumor-associated antigens (Figure 4d).

### 3.5. Therapeutic Vaccination with PeptiCRAd Cancer Vaccine Targeted to Endogenous Retroviral Antigen Shows Antitumor Efficacy

Encouraged by the marked antitumor protection in this tumor model, which is typically difficult to treat, we wished to explore further the 4T1 tumor antigenic landscape for improved therapeutic efficacy using the identified ERV (Section 3.2). The ERV antigen is a potential example of a genomic region silenced in normal tissues but transcriptionally and translationally awakened in cancer cells. For instance, according to the Aceview NCBI resource of curated cDNA sequences, LOC72520 cDNA has been found in spontaneous mouse mammary tumors that metastasized to the lung [34]. Moreover, these types of normally non-coding genomic regions could potentially be recurring in many tumors, and as such may actually be shared antigens with high immunogenic potential. We therefore explored the therapeutic potential of the most confident example found here, the FYLPTIRAV peptide from ERV LOC72520 and the TYVAGDTQV peptide.

To this end, we designed a similar therapeutic experiment (similar treatment schedule as in Figure 4a) to study the therapeutic potential of these peptides in a therapeutic vaccination setting. PeptiCRAd was used as the vaccine platform and was complexed with the FYLPTIRAV and TYVAGDTQV peptides separately (ERV-PeptiCRAd). Mice receiving ERV-PeptiCRAd showed the highest tumor protection, which was statistically significant in comparison to the virus control (*p* < 0.001) (Figure 5a). Combination treatment with checkpoint inhibitor did not increase the level of protection, recapitulating our finding using the TAA-PeptiCRAd. The identified antigen induced weak, systemic immunological responses, as re-stimulation of splenocytes with these peptides showed measurable IFN-gamma production in vaccinated mice (Figure 5b).

## 4. Discussion

Recent technological advancements in mass spectrometry have made possible the direct immunopeptidome profiling and the identification of naturally presented T-cell epitopes from tumor material with high confidence. Here we identify tumor antigens using state-of-the-art mass spectrometry and further explore their potential as cancer vaccines in a prophylactic and therapeutic setting. The depth and quality of our immunopeptidome analysis are comparable to published immunopeptidome datasets on self-antigens in tumor cell lines and normal tissues sharing similar MHC-I restriction [19,20]. Of note, three of the peptides we described here were identical to HLA-I ligands recently discovered in an immunopeptidome study on human TNBC [35]. We observed that the majority of the MHC-I ligands’ source proteins produced a single MHC-I ligand, with very few exceptions showing a wide presentation coverage. Thus, our study is in line with recent immunopeptidome analyses and offers a quality addition to the known MHC-I ligand data sets.

The advantage of utilizing direct identification for tumor antigen discovery is that the approach unveils the most clinically relevant target candidates for vaccine design. Recent successful clinical trials have validated this approach to produce candidate targets amenable as cancer vaccines with clinical benefit [8,36]. The obvious benefit of using tumor-associated antigens is that they can be ‘shared’ among patients. Of note, the utilization of ‘shared’ tumor-associated antigens does not restrict the personalization of immunotherapies, as the choice of antigens may be based on the characteristics of the patient’s individual tumor (e.g., expression of specific tumor antigens or surrogate biomarkers). Moreover, additional cancer vaccination strategies based on neoepitopes from individual patients can be designed to later complement the personalized treatment. Candidate libraries of ‘shared’ antigens have recently been created for tumor types with low neoantigen load and successfully used as ‘off-the-shelf’ vaccines in clinical trials including glioblastoma [8] and TNBC [37]. However, care should be taken when choosing the antigen target not to induce any severe on-target off-tumor effects, leading to development of autoimmune reactions.

An exciting addition to the tumor antigen landscape is tumor-specific antigens derived from normally non-coding regions. Like tumor-associated antigens, these antigens could be shared between patients, be highly immunogenic, and present clinical targets for cancer T-cell recognition [11,38]. Our results warrant further studies on the prevalence, abundance, and most importantly MHC-I restricted presentation of antigens derived from annotated non-coding regions in human tumors and highlight the usefulness of accurate mass spectrometry-based approaches for their discovery.

We show here that cancer vaccines targeting multiple tumor antigens control tumor growth in aggressive mouse model of TNBC. Importantly, we show here for the first time that control of established tumors is achievable by integrating immunopeptidomic-based antigen discovery with the PeptiCRAd oncolytic adenovirus platform. This establishes a pipeline for improved oncolytic cancer immunotherapy by potentiating further personalization of oncolytic vaccine responses. The qualitative differences in the infiltrated T-cells as well presence of antigen-specific T-cell response in the therapeutic experiment suggest the vaccination protocols succeeded in triggering antitumor immunity. Modulation of the tumor microenvironment towards a more inflammatory state with the use of an oncolytic virus-based vaccine platform might have further sensitized TNBC cells to immune therapy [32,33]. However, due to the relatively modest systemic antigen-specific responses in vaccinated mice, we additionally evaluated spontaneous immune responses against the dominant CD8+ epitope in BALB/c models (gp70/AH1) [39] in the therapeutic vaccination. Response against gp70 was detected in PeptiCRAd-vaccinated mice, but not in mock-treated mice. Thus, despite the association of the vaccine-induced antigen-specific responses with anti-tumor activity, we cannot rule out the possibility of spontaneous immune responses playing an additional role in tumor control. Additionally, the anti-tumor effect may yet be improved, as the biological effects in the current study were modest despite obtaining statistical differences between the control and treatment groups. We anticipate further combination treatment, counteracting more efficiently the suppressive tumor microenvironment, may be required to further maintain the vaccine effect. Tim-3 blockade would be especially intriguing in the context of triple-negative breast cancer as it additionally dampens the myeloid-derived suppressors [40]; however, exploring this in our model was out of the scope of the current study.

## 5. Conclusions

Our study highlights the feasibility of identifying clinically relevant tumor antigens with direct immunopeptidomic profiling and supports their utilization as vaccines for moderately immunogenic cancer types. Furthermore, we present a pipeline integrating antigen discovery with a therapeutic cancer vaccine platform for improved personalized oncolytic cancer immunotherapy. Our dataset provides tools for further proof-of-concept testing of tumor antigen vaccination protocols in combination with other immunotherapies.

## Figures and Tables

**Figure 1 cancers-13-03408-f001:**
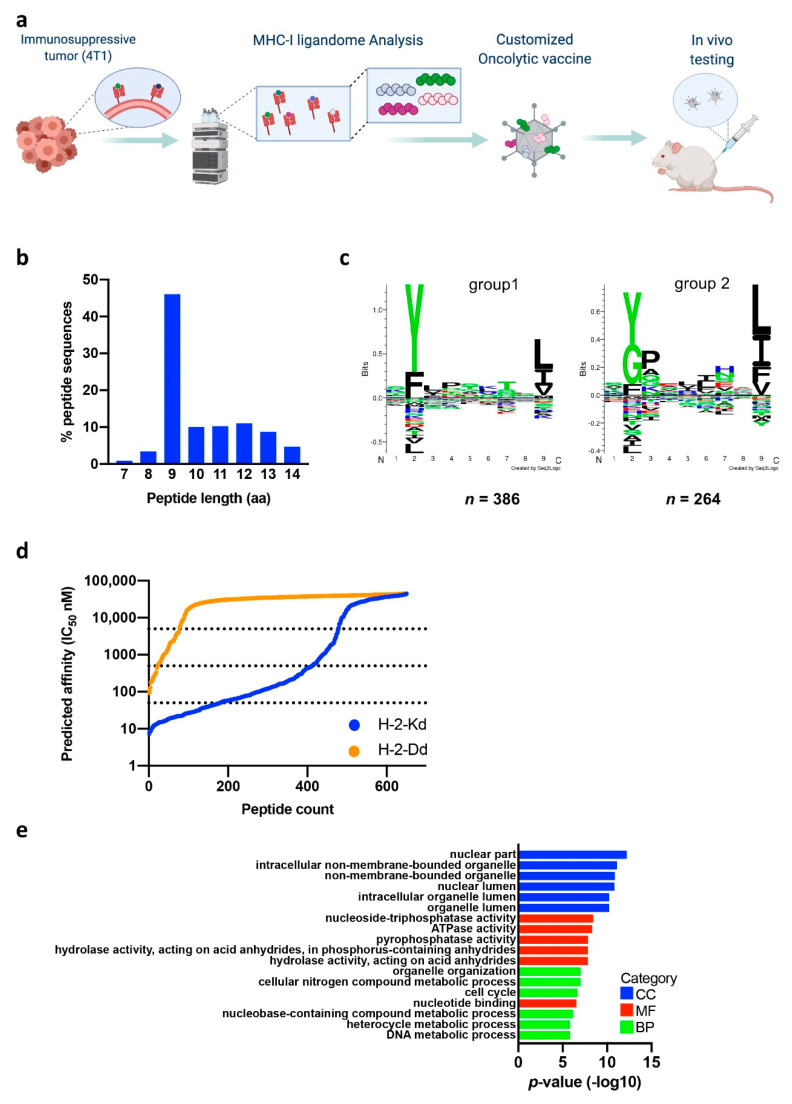
Characteristics of identified peptide ligands. (**a**) Overall workflow of the study. (**b**) Length distribution of the identified peptides. (**c**) Motif analysis of 9 mers. The 9 mers show reduced amino acid complexity at anchor positions characteristic of MHC-I peptide binding motifs expressed by the cell line. (**d**) MHC-I binding affinity prediction of 9 mers. Of the identified 9 mers, 67% can be assigned to H-2K^d^ or H-2D^d^ and thus are considered as MHC-I ligand. (**e**) GO analysis of the MHC-I ligand source proteins. Most overrepresented biological processes (BP), molecular function (MF), and cellular component (CC) of the MHC-I source proteins (*p*-values, Fisher’s exact test with FDR correction) are presented.

**Figure 2 cancers-13-03408-f002:**
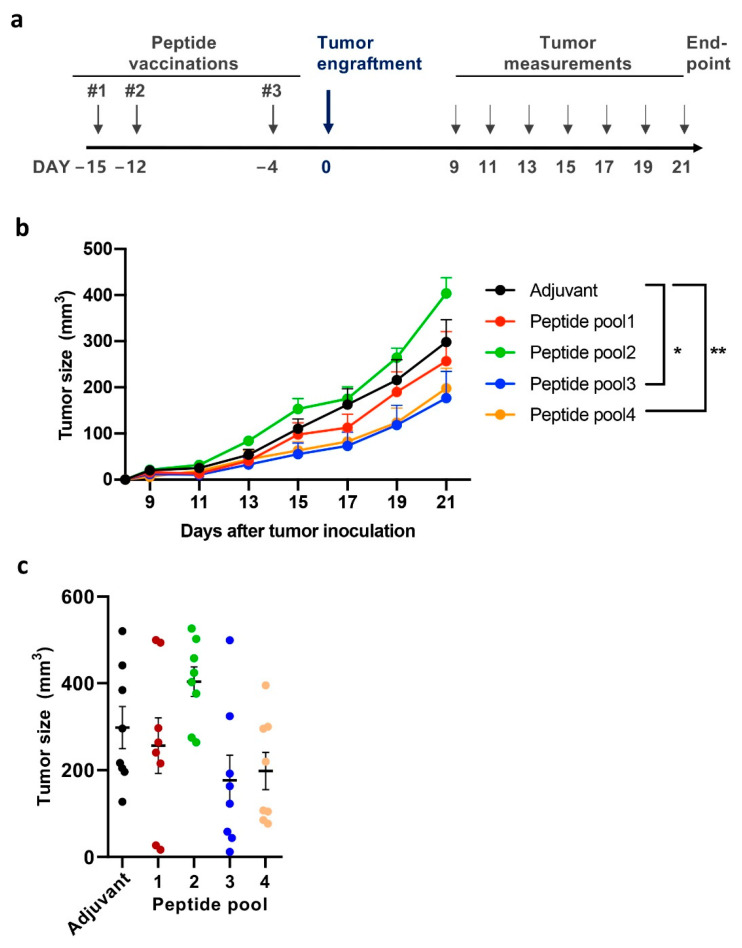
Tumor control potential of tumor-associated antigens. (**a**) Animal experiment treatment schedule. Balb/c mice were immunized with an adjuvant (poly I:C) in combination with irrelevant peptide (SIINFEKL that binds to H-2Kb and not the BALB/c expressed H-2Kd or H-2Dd (Adjuvant group) or a pool of 5-6 peptides (groups: Peptide 1, 2, 3 and 4; 25 ug each peptide per pool)). Following three peptide vaccinations, 4T1 tumors were engrafted (300,000 cells/tumor). Tumor measurements were initiated once the tumors became palpable. (**b**) Tumor growth curves for immunization groups. Peptide pool 3 and Peptide pool 4 group vaccinations control tumor growth in the aggressive 4T1 model, as compared to mock immunization. Statistical analysis of tumor growth curves was performed using two-way Anova, * *p* < 0.05, ** *p* < 0.01. The error bars show SEM, *n* = 8. (**c**) Individual tumor sizes at the endpoint (day 21 after tumor engraftment) are presented.

**Figure 3 cancers-13-03408-f003:**
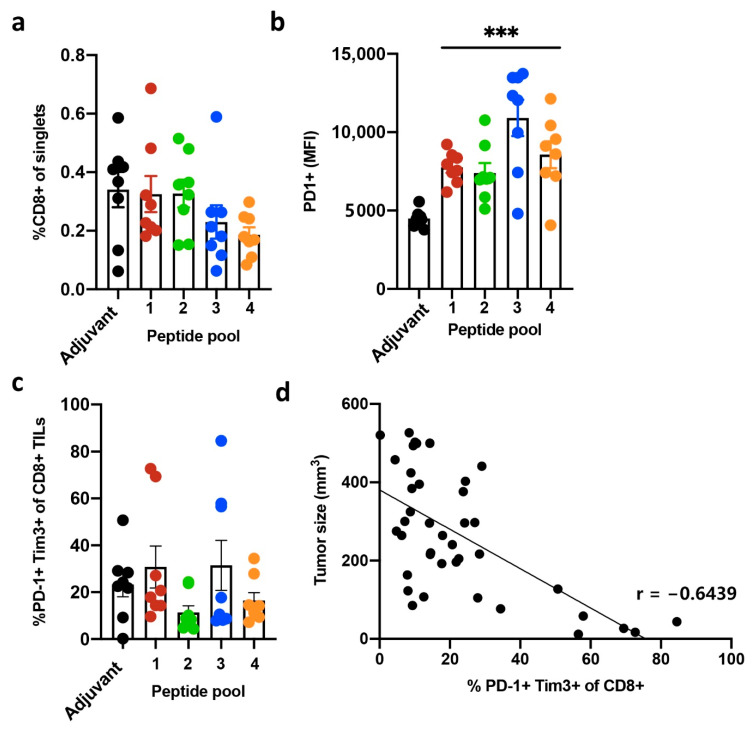
Flow cytometric analysis of tumor-infiltrating CD8+ T-cells. Dots represents analysis of individual tumors, and bars represent the mean of the group. (**a**) Percent of CD8+ cells of the singlets. (**b**) Median expression of PD-1 in CD8+ T-cells. Statistical analysis was performed using unpaired, two-tailed *t*-test, *** *p* < 0.006. (**c**) Percent PD-1+ and Tim3+ double-positive CD8+ cells. (**d**) Correlation analysis of tumor size and percentage of PD-1+ Tim3+ CD8+ cells. The tumor size (day 21) negatively correlates with percentage of infiltrated PD-1+ Tim3+ double positive CD8+ cells. Pearson correlation, r = −0.6439.

**Figure 4 cancers-13-03408-f004:**
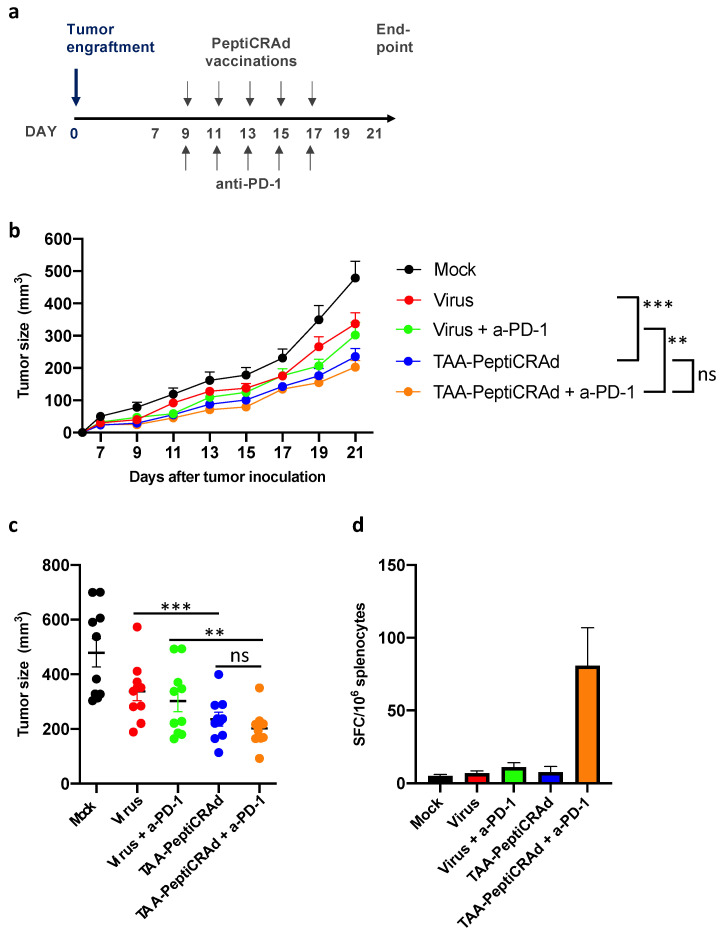
Therapeutic vaccination with PeptiCRAd targeted to tumor-associated antigen induces anti-tumor activity in 4T1 model. (**a**) Animal experiment treatment schedule. Balb/c mice were engrafted with 4T1 tumors (300,000 cells/tumor). Mice were assigned to treatment groups at day 7 blindly. Treatments were initiated at day 9 once the tumors were palpable. Each mouse received 5 injections of virus or PeptiCRAd (i.t.) and 5 injections of anti-PD-1 (i.p.) or PBS (mock injections). (**b**) Tumor growth curves for treatment groups. Treatment groups: mock injected (Mock); PeptiCRAd virus backbone without peptide loading (Virus); PeptiCRAd loaded with tumor-associated antigens (TAA-PeptiCRAd) with or without anti-PD-1 (a-PD-1). The immunopeptidomics discovered peptides FYLETQQQI, SYHPALNAI, and KYQAVTATL originating from Brap, Birc6, and Rpl13a source proteins, respectively, were used as antigens for the TAA-PeptiCRAd, and poly-lysine-tail was added to the endogenous peptides to allow PeptiCRAd complex to formulate. Statistical analysis of tumor growth curves was performed using two-way Anova with Tukey’s multiple comparison, ** *p* < 0.05, *** *p* < 0.01. The error bars show SEM, *n* = 10. (**c**) Individual tumor sizes at the endpoint (day 21 after tumor engraftment) are presented. (**d**) Immune responses against tumor-associated antigens at endpoint. Splenocytes were harvested at the end of the therapeutic experiment from 3 mice from each treatment group and stimulated with the tumor-associated antigens peptides for 2 days. Inductions of peptide specific T-cell responses were analyzed by interferon-gamma enzyme-linked immunospot (ELISPOT). Spot forming colonies (SFC) were read and quantified at ImmunoSpot (CTL Europe).

**Figure 5 cancers-13-03408-f005:**
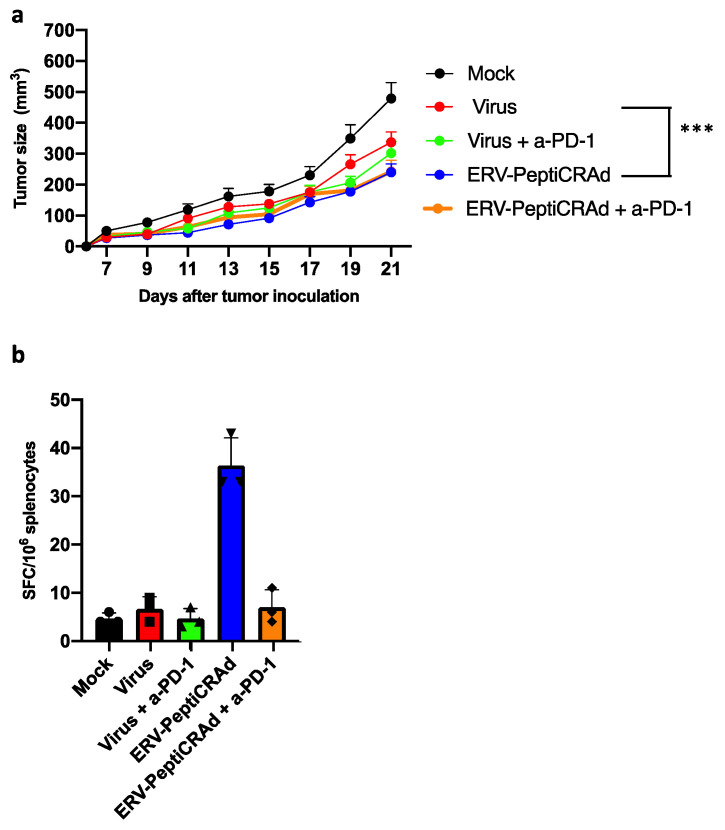
Therapeutic vaccination with PeptiCRAd targeted to endogenous retrovirus induces anti-tumor activity in the 4T1 model. (**a**) The animal experiment treatment schedule and tumor engraftment were the same as in Figure 4 experiment. Treatment groups: mock injected (Mock); PeptiCRAd virus backbone without peptide loading (Virus); PeptiCRAd loaded with endogenous retrovirus antigens (ERV-PeptiCRAd) with or without anti-PD-1 (a-PD-1). Statistical analysis of tumor growth curves was performed using two-way Anova with Tukey’s multiple comparison, *** *p* < 0.01. The error bars show SEM, *n* = 10. (**b**) Immune responses against antigens at endpoint were analyzed as in Figure 4 using ELISPOT. Spot forming colonies (SFC) were read and quantified at ImmunoSpot (CTL Europe).

**Table 1 cancers-13-03408-t001:** Peptide characteristics and peptide pools used for therapeutic vaccination.

#	Peptide	UniProt ID	Protein Description	Short Name	H-2Kd IC50 (nM)	Known MHC-I Lgand (IEDB) *	MHC-I Ligand in Normal Tissue **
**Peptide pool1**							
#1	RYLPAPTAL	Q9JL70	Fanconi anemia group A protein homolog	Fanca	13.44	Yes	-
#2	FYITSRTQF	F8WI90	Tyrosine-protein kinase	Scr	15.05	Yes	Yes
#3	SYFPEITHI	B1ASP2	Tyrosine-protein kinase	Jak1	21.79	Yes	-
#4	FYLETQQQI	Q99MP8	BRCA1-associated protein	Brap	29.37	-	-
#5	NYVPGKFTV	E9PXX8	Metastasis-associated in colon cancer 1	Macc1	59.45	-	-
**Peptide pool2**							
#6	EYVHTKNFI	H7BXB1	Casein kinase I isoform alpha	Csnk1a1	65.81	-	-
#7	NYQDTIGRL	A0A0A6YWC8	Vimentin	Vim	470.08	Yes	Yes
#8	KYLATLETL	B1ASP2	Tyrosine-protein kinase	Jak1	13.72	Yes	Yes
#9	YFISSTTRI	A0A0A0MQ80	Spermatogenesis-associated protein 5	Spata5	29.64	Yes	-
#10	SYLKSELGL	A2AQD5	Sperm-specific antigen 2 homolog	Ssfa2	121.09	-	-
**Peptide pool3**							
#11	SYHPALNAI	S4R1L5	Baculoviral IAP repeat-containing protein 6	Birc6	9.9	Yes	-
#12	SYYAVAHAV	A0A0R4J170	Transcription activator BRG1	Smarca4	11.16	-	-
#13	AYKAVLNYL	D3YXN3	Testis-expressed protein 30	Tex30	43.77	Yes	-
#14	EYVANLTEL	A0A0R4J170	Transcription activator BRG1	Smarca4	100.38	Yes	-
#15	KYSAQIEDL	B1AUF1	Ski oncogene	Ski	443.63	-	-
**Peptide pool4**							
#16	EYIHSKNFI	Q9JMK2	Casein kinase I isoform epsilon	Csnk1e	26.77	Yes	-
#17	KYQAVTATL	P19253	60S ribosomal protein L13a	Rpl13a	14.51	Yes	Yes
#18	KYQEALDVI	Q8BWZ3	N-alpha-acetyltransferase 25, NatB auxiliary subunit UV excision repair protein RAD23 homolog B, HR23B	Naa25	31.04	Yes	-
#19	SYENMVTEI	P54728	mHR23B	Rad23b	9.65	Yes	-
#20	SYKPIVEYI	Q8C1B7	Septin 11	Sept11	45.35	Ye	-
#21	TYVPIAQQV	A2APB8	Targeting protein for Xklp2	Tpx2	90.44	Yes	-

* IEDB (Immune Epitope Database), ** Described as HLA-I ligand in the ‘HLA Ligand Atlas’.

## Data Availability

The mass spectrometric raw datasets generated during the current study are available in the ProteomeXchange Consortium via the PRIDE partner repository with the dataset identifier PXD016112.

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
