# Peer review of "Therapeutic Cancer Vaccination with Immunopeptidomics-Discovered Antigens Confers Protective Antitumor Efficacy"

_cancers, 2021, doi:10.3390/cancers13143408_

Round 1

Reviewer 1 Report

The authors have addressed my concerns. A minor last comment that can be fixed during the proof step: the charge states of the annotated fragments in the MSMS spectra should be added (together with other losses if included).

Author Response

Response: We thank the reviewer for excellent comments throughout the review process. The comments have clearly improved the quality of the manuscript.

The charge states of the annotated fragments will be provided in the Supplementary data.

Reviewer 2 Report

I think the authors of the paper took into account all the criticism, revised the article making it ready for publication.

Author Response

Response: We thank the reviewer for excellent comments throughout the review process. The comments have clearly improved the quality of the manuscript.

Reviewer 3 Report

The authors present an interesting study with a novel set of immunotherapeutic targets presented in an oncolytic/peptide vaccine setting. However, significant details are missing from the manuscript which make it difficult to understand how some of the experiments were performed. Please address the following issues:

  1. More information is required to interpret Figure 1D. What is the nM cutoff for “high-confidence binders”?

  1. Besides peptide pool 4, it is not clear how the peptides in the other pools were prioritized and grouped. Regardless of how this was done, it needs to be included in the text somewhere.

  1. Line 341 – “An irrelevant peptide plus adjuvant was used 341 as mock control.” What is the irrelevant peptide?

  1. As the primary goal of the paper is to demonstrate proof of concept for their peptide discovery pipeline, the authors need to include some evidence that CD8 T cell responses are generated to the peptides (peptide pools 1-4). An in vitro peptide restimulation assay followed by cytokine measurement by flow or ELISA or ELISPOT would be appropriate. Deconvoluting the pools would be even better to measure against individual peptides.

  1. Which group of antigens were chosen for the TAA-PeptiCRAd experiment (Fig 4)?

  1. What additional advantage does the formulation of peptide in “peptiCRAd” offer in terms of vaccination poly IC? Measurement of T cell responses would be appropriate.

  1. Line 452 reference to wrong figure? “To this end, we designed a similar therapeutic experiment (Figure 4A) to study the therapeutic potential of these peptides in a therapeutic setting”

  1. Please provide more information about how the statistics were done… Based on the data in supplementary data figure 4, how is this p value achieved? It should be acknowledged that even if that is statistically significant, the biological effect is rather modest.

  1. In Figure 5B, please show individual mice on graph. Why is the T cell response to ERV PeptiCRAd worse with antiPD1?

  1. Many of the antigens chosen are ubiquitously expressed self antigens to which tolerance mechanisms are in place. Can the authors provide some discussion about the challenges associated with this type of antigen? Do the authors expect that if tolerance is broken that there may be autoimmunity?

  1. The categorization of mice into either responders or nonresponders is very confusing because it is also being applied to the control group.

Additionally, there are several instances where there is red text that is out of place. Perhaps

this a formatting problem when the doc was converted to a pdf? I have indicated these below:

Line 118 – “One” out of place

Line 139 – “For the discovery of non-canonical peptides (ERV antigens), we performed a re-final raw file…”

Line 308- “but one two belonged to theoretically non-coding genomic regions”

Line 309 – “TOne, this peptide,he FYLPTIRAV peptide (confidence score |log prob|=4.75) from  hypothetical…”

Line 312-314 “ We further validated the identification of FYLPTIRAV these peptides with a synthetic peptideones, the mirror plots showing endogenous peptides and respective synthetic peptide is presented as Supplementary data file 2”

Line 443: “TBoth the ERV antigen is a and pseudogene antigens are potential examples”

Sentence unclear/non-informative: lines 156-157 – “Unsupervised alignment and clustering of input 156 peptides (9mers, 10mers or 11mers separately) as 1-5 clusters and respective motif lengths 157 was carried out using default settings.”

Lines 198-201 in red are repeated in lines 202-204.

Author Response

Response: We thank the reviewer for the interest towards our study and for excellent and comments and through review. The questions have clearly improved the quality of the manuscript. Please find out point-by-point response to each of the addressed point below.

  1. More information is required to interpret Figure 1D. What is the nM cutoff for “high-confidence binders”?

Response: The cutoff we used for defining high-confidence binders is 500nM IC50. This is commonly used cutoff value for predicted peptide binding to respective MHC-I molecule. We added this information in the main text, and the chapter reads now:

‘We next aimed to identify the high-confidence MHC-I ligands among the 9mers. Prediction of the peptide binding affinity to MHC-I showed 67% of the 9mers bind either H-2 Kd or H-2 Dd and can thus be considered as MHC-I ligands (using the <500nM IC50 affinity value as cutoff for MHC-I binding, Figure 1D). The vast majority of these ligands showed clear preference to H-2Kd and 41% of the H-2Kd ligands bound H-2Kd with high affinity (<50nM IC50, Supplementary data file1).’

  1. Besides peptide pool 4, it is not clear how the peptides in the other pools were prioritized and grouped. Regardless of how this was done, it needs to be included in the text somewhere.

Response: The selection of peptides to be tested in further experiments is challenging because currently there exists no harmonised criteria or characteristics in the field that could be used to identify the most promising antigen candidates. This is related to the fact that predicting peptide immunogenicity is currently very challenging. For these reasons we wished to explore our in-house developed tool for peptide selection. Our primary criteria for selecting peptides for the other pools was high (predicted) peptide affinity for MHC-I. We were initially interested in including peptides derived from source proteins that could be regarded as cancer testis antigens or are known to have a function in cancer biology (eg. Ski oncogene). This strategy to select peptides does not, to our knowledge, differ from the many published protocols and indeed very few articles reveal their criteria for peptide selection in more detail. Our primary focus with the current study was to establish the overall workflow as described in Supplementary Figure 1, and we aim to improve our peptide selection strategy in the follow-up studies. 

  1. Line 341 – “An irrelevant peptide plus adjuvant was used 341 as mock control.” What is the irrelevant peptide?

Response: The irrelevant peptide refers to peptide that cannot bind to the Balb/c mouse MHC-I, in this case either H-2-Kd, H-2Dd, or H-2Ld. We chose a well known peptide SIINFEKL which is commonly used in immunological studies in the context of C57BL/6 mouse strain and binds to H-2Kb with high affinity. This information is included in the Figure 2 legend.

  1. As the primary goal of the paper is to demonstrate proof of concept for their peptide discovery pipeline, the authors need to include some evidence that CD8 T cell responses are generated to the peptides (peptide pools 1-4). An in vitro peptide restimulation assay followed by cytokine measurement by flow or ELISA or ELISPOT would be appropriate. Deconvoluting the pools would be even better to measure against individual peptides.

Response: We agree with the reviewer that showing direct restimulation with ELISPOT or ELISA would further strengthen the study. We performed ELISPOT assay where we restimulate the splenocytes with the peptides and analysed IFN-gamma production to measure T-cell activation. As we utilized short peptides (9mers) in the restimulation with the ability to bind only MHC-I (and not MHC-II) we expect the responses reflect CD8 activation. We noted the immunological responses obtained from experiments conducted at our laboratory were very modest, probably partly due to biological and partly due to technical reasons. Because of this reason we analysed further the tumor samples for T-cell infiltration using flow cytometric analysis. We could indeed show that the peptide pool vaccinated mice tumors CD8 T-cells expressed higher level of PD-1 in comparison to adjuvant vaccinated mice. Additionally, peptide pool vaccinated mice had higher infiltration of PD-1+ Tim3+ CD8 cells reflecting previous antigen exposure and exhaustion. Collectively, we used these data as surrogate for the ex vivo restimulation and measure T-cell responses generated towards the peptides in vivo.

  1. Which group of antigens were chosen for the TAA-PeptiCRAd experiment (Fig 4)?

Response: The peptides FYLETQQQI, SYHPALNAI and KYQAVTATL originating from Brap, Birc6 and Rpl13a source proteins, respectively, were used in the Figure 4 experiment. These are known mouse antigens (found in IEDB antigen database) or alternatively their human ortholog is a known antigen. This information is found within the Materials and Methods section ‘Mice and animal experiment’.

  1. What additional advantage does the formulation of peptide in “peptiCRAd” offer in terms of vaccination poly IC? Measurement of T cell responses would be appropriate.

Response: We have previously shown higher infiltration of lymphocytes to tumors with the PeptiCRAd vaccination in comparison to mock treatment. Importantly, these lymphocytes are cytotoxic and antigen-specific (Ylösmäki et al., Characterization of a novel OX40 ligand and CD40 ligand-expressing oncolytic adenovirus used in the PeptiCRAd cancer vaccine platform, Mol Ther Oncolytics;  https://doi.org/10.1016/j.omto.2021.02.006).

The peptide in PeptiCRAd formulation is designed to guide antigen-specificity of the vaccine platform. We provide preliminary experimental evidence for improved antigen-specific immune response obtained using PeptiCRAd over peptide-polyI:C vaccination using known, well established antigen as new Supplementary Figure 4.

  1. Line 452 reference to wrong figure? “To this end, we designed a similar therapeutic experiment (Figure 4A) to study the therapeutic potential of these peptides in a therapeutic setting”

Response: Here our idea was to refer to the previous therapeutic experiment treatment scheme presented as Figure 4A. We have revised the sentence, and it stands now: ‘To this end, we designed a similar therapeutic experiment (similar treatment schedule as in Figure 4A) to study the therapeutic potential of these peptides in a therapeutic setting.’

  1. Please provide more information about how the statistics were done… Based on the data in supplementary data figure 4, how is this p value achieved? It should be acknowledged that even if that is statistically significant, the biological effect is rather modest.

Response: Statistical analysis was performed using GraphPad Prism 6.0 software. For tumor growth curve analysis 2-way ANOVA with Tukey’s multiple comparison was chosen as statistical method. The dataset for the comparison was the complete growth curves not merely the data at the endpoint. As input to the statistical analysis individual absolute tumor volume values were used.

Only selected comparisons were highlighted in the Figure 4 and Figure 5. There is statistical difference between the groups:

Tukey's multiple comparisons test

Mean Diff,

95,00% CI of diff,

Significant?

Summary

Adjusted P Value

Mock vs. Virus

48,9

22,57 to 75,24

Yes

****

<0,0001

Mock vs. TAA-PeptiCRAd

88,36

62,03 to 114,7

Yes

****

<0,0001

Mock vs. ERV-PeptiCRAd

90,28

63,94 to 116,6

Yes

****

<0,0001

Mock vs. Virus + PD-1

65,48

39,14 to 91,81

Yes

****

<0,0001

Mock vs. TAA-PeptiCRAd + PD-1

100,9

74,60 to 127,3

Yes

****

<0,0001

Mock vs. ERV-PeptiCRAd + PD-1

78,94

52,60 to 105,3

Yes

****

<0,0001

Virus vs. TAA-PeptiCRAd

39,46

13,13 to 65,80

Yes

***

0,0002

Virus vs. ERV-PeptiCRAd-PeptiCRAd

41,38

15,04 to 67,71

Yes

****

<0,0001

Virus vs. TAA-PeptiCRAd + PD-1

52,03

25,69 to 78,37

Yes

****

<0,0001

Virus vs. ERV-PeptiCRAd + PD-1

30,04

3,703 to 56,38

Yes

*

0,013

Virus + PD-1 vs. TAA-PeptiCRAd + PD-1

35,45

9,118 to 61,79

Yes

**

0,0012

We have included more information on the statistical difference between the treatment groups in the Supplementary Figure 5 figure legend. We agree that despite obtaining statistical significance the biological effect could still be further improved. We have added this point in the last paragraph of the discussion.

  1. In Figure 5B, please show individual mice on graph. Why is the T cell response to ERV PeptiCRAd worse with antiPD1?

Response: We believe one explanation to the apparent discrepancy in our results is partly biological partly technical. The PeptiCRAd vaccine platform is relatively novel and as such immunologically not yet fully characterized. However, we anticipated it is possible we do not obtain very robust differences in the immunological responses between PeptiCRAd and PeptiCRAD + anti-PD-1 group based on our previous experience. In our experience in the 4T1 mouse model the antigen-specific immunological responses are in general much weaker for the tumor antigens in comparison to the immunodominant antigen (gp70/AH-1). Furthermore, we were able to include only three randomly chosen mice from each treatment group for the immunogenicity test. The responses of these individual mice is now included as part of Figure 5B. It may appear that as individual mouse responses are variable, and the sensitivity of the present experiment is not optimal, even small changes may appear pronounced in the final result. Additionally, with the current experimentation we cannot rule out the possibility of antigen or epitope spreading playing a role in the treatment responses. Clearly, thorough experimentation would be required to understand this apparent discrepancy in more detail.

  1. Many of the antigens chosen are ubiquitously expressed self antigens to which tolerance mechanisms are in place. Can the authors provide some discussion about the challenges associated with this type of antigen? Do the authors expect that if tolerance is broken that there may be autoimmunity?

Response: We agree with the reviewer that the selection of some antigens, ie the one with ubiquitous expression, may have not been optimal and we aim to improve our antigen selection in future studies. To this end, neoantigens derived from tumor mutations or even cancer testis antigens expressed ‘normally’ solely in immune privileged organs would offer more tumor specific alternative. However, these types of antigens may not be viable options for cancer types expressing or presenting low amount of these antigens. Hence, the use of tumor associated antigens may be justified for certain tumor types.

Our current study does not take a stand on the potential risks related to therapeutic cancer vaccines. However, we recognize the risk of developing autoimmune reactions as long-term ‘side-effects’. For instance, development of uveitis has been reported in patients treated with adoptive TILs therapy (melanoma-specific CD8+ cells). Additionally, severe autoimmunity related issues among breast cancer patients has been observed with the use of Trastuzumab therapeutic monoclonal antibody. Trastuzumab has high incidence for cardiotoxicity due to HER2 targeting in heart tissue. However, this is not relevant for triple-negative-breast cancer as the tumor type lacks HER2 expression and thus HER2 based treatment options are not desirable. We agree with the reviewer that care should be taken when choosing the antigen target and likewise vaccination schedule and dosing should be designed to minimize any therapy related on-target off-tumor effects. Short discussion on this matter is now included in the manuscript discussion, at the end of second paragraph.

  1. The categorization of mice into either responders or nonresponders is very confusing because it is also being applied to the control group.

Response: individual mice were scored as responders (at least 30% decrease in tumor volume in comparison to the median tumor volume of the adjuvant group) or non-responder (at least 20% increase in tumor volume in comparison to the median tumor volume of the mock group) at the study endpoint. Hence, the categorization was predetermined at the beginning of the study using unbiased criteria instead of setting the cutoff at the end of the experiment. Using this criteria eg. two mice fall just below the ‘responder’ threshold in the ‘Mock group’ of the therapeutic vaccination experiment (Supplementary Figure 5). We removed the categorization of the control groups in the Supplementary Figure 2 and Supplementary Figure 5 for the revised manuscript version.

Additionally, there are several instances where there is red text that is out of place. Perhaps

this a formatting problem when the doc was converted to a pdf? I have indicated these below:

Line 118 – “One” out of place

Line 139 – “For the discovery of non-canonical peptides (ERV antigens), we performed a re-final raw file…”

Line 308- “but one two belonged to theoretically non-coding genomic regions”

Line 309 – “TOne, this peptide,he FYLPTIRAV peptide (confidence score |log prob|=4.75) from  hypothetical…”

Line 312-314 “ We further validated the identification of FYLPTIRAV these peptides with a synthetic peptideones, the mirror plots showing endogenous peptides and respective synthetic peptide is presented as Supplementary data file 2”

Line 443: “TBoth the ERV antigen is a and pseudogene antigens are potential examples”

Sentence unclear/non-informative: lines 156-157 – “Unsupervised alignment and clustering of input 156 peptides (9mers, 10mers or 11mers separately) as 1-5 clusters and respective motif lengths 157 was carried out using default settings.”

Lines 198-201 in red are repeated in lines 202-204.

Response: We believe these may indeed present formatting issues, and now provide the revised manuscript also as ‘clean’ pdf version.

Round 2

Reviewer 3 Report

Thank you for your edits and comments.

Specifically with respect to the comment below, can the authors please add the specific peptides/antigens to the main text and or Figure legend? It is difficult to associate from the Materials and methods.

  1. Which group of antigens were chosen for the TAA-PeptiCRAd experiment (Fig 4)?

Response: The peptides FYLETQQQI, SYHPALNAI and KYQAVTATL originating from Brap, Birc6 and Rpl13a source proteins, respectively, were used in the Figure 4 experiment. These are known mouse antigens (found in IEDB antigen database) or alternatively their human ortholog is a known antigen. This information is found within the Materials and Methods section ‘Mice and animal experiment’.

Author Response

Thank you for your edits and comments.

Specifically with respect to the comment below, can the authors please add the specific peptides/antigens to the main text and or Figure legend? It is difficult to associate from the Materials and methods.

  1. Which group of antigens were chosen for the TAA-PeptiCRAd experiment (Fig 4)?

Response: The peptides FYLETQQQI, SYHPALNAI and KYQAVTATL originating from Brap, Birc6 and Rpl13a source proteins, respectively, were used in the Figure 4 experiment. These are known mouse antigens (found in IEDB antigen database) or alternatively their human ortholog is a known antigen. This information is found within the Materials and Methods section ‘Mice and animal experiment’.

Response: Thank you for the the further comment. The suggested addition is now made both in the main text and the Figure 4 legend.

This manuscript is a resubmission of an earlier submission. The following is a list of the peer review reports and author responses from that submission.

Round 1

Reviewer 1 Report

The manuscript describes an approach for cancer vaccination that includes identification of immunogenic peptides, which characterize malignant cells. The approach also involves two variants of delivery of these identified peptides into the experimental tumor. First, the peptides can be delivered by themselves and second, they can be delivered being complexed with an oncolytic adenovirus carrier. First variant of delivery helped in identifying pools of peptides that work prophylactically being injected before tumor implantation.  Almost all selected peptide pools were able to slow down consequent tumor growth. The second variant of the delivery demonstrated that identified peptides after complexing with viral carrier were able to work therapeutically by slowing down tumor growth. Authors of the study also identified one particular peptide that alone can work therapeutically after being complexed with carrier adenovirus.

I think that the study is important for the field and very interesting. It is also performed with great skill and perseverance. In addition, the study is well designed and well presented. Authors of the manuscript demonstrated benefits of complexing antigenic peptides and oncolytic viruses for antitumor therapy. Their work also presents experimental and bioinformatic peptide-neoantigen identification scheme for a particular tumor that can be scaled up for any malignancy. There are minor problems and revision suggestions.

  • I think that addition of flow charts with schematic illustration(s) of logical blocks of sections: Proteomics database search page 3, and section: In silico analysis of the MHC-I peptides, page 4, is going to be beneficial for the paper. Manuscript authors might choose to place the flow-chart(s) in the main text or as a supplementary figure(s). Without such illustration(s), understanding of the steps of bioinformatic analysis is a time-consuming process.
  • Image of supplementary Figure 4 is duplicating Figure 4 in the main text.
  • The text of the abstract is not always clear and can be improved. Some words are too repetitive, and this repetition is confusing. The word “Immunopedidome” occurs three times, and it supplemented with one additional “immunopeptidomics”. I think that clarity of the text is going to be improved by using synonym at least once.
  • Some of the long sentences in the abstract are overloaded with information and can be broken down into shorter sentences for clarity. For example, the sentence:” We show the clinical…”(Page 1, line 28 and 29) is difficult for comprehension because of its length.
  • Page 1, line 34. “Oncolytic cancer immunotherapy” is excessive, it can be “oncolytic immunotherapy”, the word “cancer” is not needed, because “oncolytic” is already referred to malignancy.

Author Response

The manuscript describes an approach for cancer vaccination that includes identification of immunogenic peptides, which characterize malignant cells. The approach also involves two variants of delivery of these identified peptides into the experimental tumor. First, the peptides can be delivered by themselves and second, they can be delivered being complexed with an oncolytic adenovirus carrier. First variant of delivery helped in identifying pools of peptides that work prophylactically being injected before tumor implantation.  Almost all selected peptide pools were able to slow down consequent tumor growth. The second variant of the delivery demonstrated that identified peptides after complexing with viral carrier were able to work therapeutically by slowing down tumor growth. Authors of the study also identified one particular peptide that alone can work therapeutically after being complexed with carrier adenovirus.

I think that the study is important for the field and very interesting. It is also performed with great skill and perseverance. In addition, the study is well designed and well presented. Authors of the manuscript demonstrated benefits of complexing antigenic peptides and oncolytic viruses for antitumor therapy. Their work also presents experimental and bioinformatic peptide-neoantigen identification scheme for a particular tumor that can be scaled up for any malignancy. There are minor problems and revision suggestions.

We thank the reviewer for the positive comments about our manuscript. Please find below our point-by-point answers to the raised questions.

I think that addition of flow charts with schematic illustration(s) of logical blocks of sections: Proteomics database search page 3, and section: In silico analysis of the MHC-I peptides, page 4, is going to be beneficial for the paper. Manuscript authors might choose to place the flow-chart(s) in the main text or as a supplementary figure(s). Without such illustration(s), understanding of the steps of bioinformatic analysis is a time-consuming process.

We thank for the comment and for the good suggestion. Indeed, an illustration describing both the experimental and bioinformatic workflow clarifies these steps. As suggested, a new scheme is drawn and presented as a supplementary figure. The Supplementary Figure 1 is updated and now describes in detail the immunopeptidome workflow (Supplementary Figure 1A) and following bioinformatics analyses (Supplementary Figure 1B). Changes in text to include these additional images is made on page 5, line 242 and 248. Accordingly, previous Supplementary Figure 1 is now Supplementary Figure 1C.  

Image of supplementary Figure 4 is duplicating Figure 4 in the main text.

We thank the reviewer for identifying this mistake. The original Supplementary Figure 4 describing the individual mouse tumor growth curves in now presented.

The text of the abstract is not always clear and can be improved. Some words are too repetitive, and this repetition is confusing. The word “Immunopedidome” occurs three times, and it supplemented with one additional “immunopeptidomics”. I think that clarity of the text is going to be improved by using synonym at least once.

Some of the long sentences in the abstract are overloaded with information and can be broken down into shorter sentences for clarity. For example, the sentence:” We show the clinical…”(Page 1, line 28 and 29) is difficult for comprehension because of its length.

We thank for these comments and appreciate the request to make the abstract text more readable. The abstract text is now simplified and the word ‘immunopeptidome’ is deleted where it is not needed. Also the long sentence mentioned is simplified and stands now: ‘The clinical benefit and tumor control potential of the identified tumor antigens and ERV antigen was studied in a preclinical model using two vaccine platforms and therapeutic settings’ (page 1, line 27 – 29).

Page 1, line 34. “Oncolytic cancer immunotherapy” is excessive, it can be “oncolytic immunotherapy”, the word “cancer” is not needed, because “oncolytic” is already referred to malignancy.

Likewise, this change is made in the abstract to improve the text (page 1, line 32 in the current version).

Reviewer 2 Report

Peltonen et al. describe an approach for inducing in vivo antigen-specific anti-tumor responses in the 4T1 TNBC murine model using oncolytic adenovirus loaded with antigens identified by mass spectrometry based immunopeptidomics. With this pre-clinical platform, immunization with TAAs or with ERV antigens was shown to be effective, highlighting its clinical potential.

Overall, the approach is interesting, although tested on one model and with rather limited experimental validation and limited characterization of the induced immune responses. In addition, the structure of the manuscript is confusing. First antigen discovery is reported followed by in vivo validation, and then again, there is a second section on antigen discovery followed again by in vivo validation assays. It is not clear why there should be two sections on the immunopeptidomics, as both can be described right at the beginning. There is no clear explanation why the first two raw files were used for searching the standard proteome reference and why only the third MS raw file was used to search the extended reference. All (three?) raw MS files should be used for searching both the standard proteome reference and the extended proteome reference and ideally the peptide targets would be identified in all replicates. In addition, data and validations experiments are missing throughout the manuscript. The whole list of identified peptides should be provided as supplementary table, and the annotated MSMS of the identified targets that were used for the in-vivo assays should be provided with identification scores. As the peptides were identified with relatively high FDR, additional validation using synthetic peptides should be provided to support their correct identification. Do the authors have evidence that the ERV gene is expressed at the RNA level? Can the authors provide evidence of tumor killing that is mediated by purified antigen-specific T cells?

The authors should clarify if the column “identified in breast cancer patient” in table 1 refers to the peptide or to the source protein.  References should be added to the table related to the studies that reported their identification in breast cancer patients. A column indicating if the peptides are derived from cancer testis antigens or overexpressed cancer associated antigens should be added. Could the author specify if such source proteins have been shown to be normally presented in human immunopeptidome samples purified from healthy tissues?

TAAs in group 4 were selected based on similarity to viral epitopes. This should be clarified in the text and in the table. Were any of these peptides (or their similar viral epitopes) shown to be immunogenic in other studies?  

Figure 4: which TAAs were used in these experiments?

The authors should explain the unexpected results where ERV-PeptiCRAd with anti-PD1 did not induce antigen-specific immune responses and the ERV-PeptiCRAd by itself did (Figure 5b), while immunization with the TAA-peptiCRAd induced resposes only when anti PD1 was included (Figure 4b).

Line 103: SepPac should be Sep-Pak.

Line 105: please confirm that 0.11% FTA was used for the wash.

In figure 1b, the length distribution of the eluted peptides suggests a relatively high proportion of contaminating peptides (12-14 mers), this could be mentioned in the text.

Author Response

Peltonen et al. describe an approach for inducing in vivo antigen-specific anti-tumor responses in the 4T1 TNBC murine model using oncolytic adenovirus loaded with antigens identified by mass spectrometry based immunopeptidomics. With this pre-clinical platform, immunization with TAAs or with ERV antigens was shown to be effective, highlighting its clinical potential.

We thank the reviewer for the interest towards our work and for thorough inspection of the manuscript. Please find below our point-by-point answers to the raised criticism.

Overall, the approach is interesting, although tested on one model and with rather limited experimental validation and limited characterization of the induced immune responses. In addition, the structure of the manuscript is confusing. First antigen discovery is reported followed by in vivo validation, and then again, there is a second section on antigen discovery followed again by in vivo validation assays. It is not clear why there should be two sections on the immunopeptidomics, as both can be described right at the beginning. There is no clear explanation why the first two raw files were used for searching the standard proteome reference and why only the third MS raw file was used to search the extended reference. All (three?) raw MS files should be used for searching both the standard proteome reference and the extended proteome reference and ideally the peptide targets would be identified in all replicates.

We thank the reviewer for raising this point, and agree that the presented structure may have caused some confusion. The initial structure originated from chronological presentation of the data, however, we now see this was not optimal from the reader’s point of view. We have now gone through an extensive revision on the manuscript structure and as part of that exercise describe both the antigen discovery sections at the beginning, first the discovery of the tumor associated antigens using the reference database and then the ERV and pseudogene antigen using the extended database. As a result, the Supplementary Figure 5 is omitted.

A more detailed workflow illustration describing both the experimental and bioinformatics workflow is currently presented as new Supplementary figure 1A-B to better clarify the discovery pipeline. Importantly, we have now re-searched all the three original raw MS files against both the mouse standard proteome (53 378 entries) and the extended proteome (92 607 entries) allowing the ERV and pseudogene antigen discovery. The output of these searches (in total 6 search files) is deposited to the proteome archive (ProteomeXchange) and is available in dataset PXD016112. Additionally, both utilized search FASTA files are deposited to PXD016112.

We indeed identify the TAA peptides during the reference proteome search. The ERV peptide ‘FYLPTIRAV’ is identified within one of the raw files during the extended database search. This identification was found within the very stringent FDR cutoff (1%), and the specific peptide spectrum match has a Byonic |logprob| value of 4.75 (i.e. local peptide p-value=1.78 E-5). Hence the mass spectrometric identification is considered as a confident high scoring peptide.

In addition, data and validations experiments are missing throughout the manuscript. The whole list of identified peptides should be provided as supplementary table, and the annotated MSMS of the identified targets that were used for the in-vivo assays should be provided with identification scores.

The raw list of MS/MS identified peptides are deposited to ProteomeXchange and additionally provided as supplementary data sets. As requested, the peptides utilized in the in vivo experiments are listed in Supplementary data file 1. This file contains the qualitative data linked to their MSMS identification such as ‘score’, ‘Log Prob’ and ‘Mass error in ppm’ for easier evaluation of the data quality. The results of the two searches using the reference and extended databases are also provided within Supplementary data file 1. These files contain the peptide sequence (in the context of their source protein sequence), Uniprot accession, Uniprot protein name among other parameters.

As the peptides were identified with relatively high FDR, additional validation using synthetic peptides should be provided to support their correct identification.

We originally set the FDR to be relatively less stringent (FDR 5%; logprob 1.3) and this is not uncommon in the field of immunopeptidomics. However, we well understand the concern about the quality of the initial discovery step. To respond to the raised issue we now provide the requested validation with synthetic peptides for the MSMS identified peptides utilized in the in vivo studies as supplemental data file (Supplementary data file 2). These spectra support the MS/MS identification of the endogenous peptides. This additional validation is also pointed out in the main text in page 8 line 320.

Do the authors have evidence that the ERV gene is expressed at the RNA level?

Currently, there exists great potential to applying ERV-derived peptides as therapeutic antigens in immunotherapy and more studies on their direct identification and on their impact on tumor regression is of utmost importance. We actually originally identified the ERV transcript in our RNAseq data, hence we have evidence for the ERV gene that it is indeed transcribed and present at the RNA level (data not shown). In the current manuscript we focus on the immunopeptidome and show the ERV peptide is also MHC-I presented. As previously discussed, the ERV peptide FYLPTIRAV identification was confident and the data presented in the mirror plot further support this identification. Collectively, this evidence provides extremely strong experimental validation for the presence and the MHC-I presentation of the ERV antigen.

Can the authors provide evidence of tumor killing that is mediated by purified antigen-specific T cells?

We thank for the relevant question and, indeed, providing this validation is on our ‘wish-list’. In the manuscript we provide experimental evidence that the peptides chosen for the cancer vaccines are naturally presented. Additionally, the peptides induce modest immunogenic response when splenocytes from immunized mice are stimulated ex vivo. We show the peptide vaccination and therapeutic vaccination using the PeptiCRAd vaccine platform is associated with improved tumor control in comparison to mock vaccination (adjuvant) or oncolytic virus only.

In order to show antigen-specific tumor killing for instance a co-culture assay with tumor cells as target and purified antigen-specific T-cells as effectors would be needed. Custom-produced tetramers are requisite for the purification of the antigen-specific T-cells. Currently, we unfortunately do not have the access for these custom-made reagents as these are more limited for the mouse model MHC-I class (H-2-Kd) and are not available for us at the moment. However, we very much understand the relevance of this question and the need to pinpoint the specificity of the anti-tumor responses. Currently we, unfortunately, may solely prove an association between the peptide vaccination and tumor control.

The authors should clarify if the column “identified in breast cancer patient” in table 1 refers to the peptide or to the source protein.  References should be added to the table related to the studies that reported their identification in breast cancer patients. A column indicating if the peptides are derived from cancer testis antigens or overexpressed cancer associated antigens should be added. Could the author specify if such source proteins have been shown to be normally presented in human immunopeptidome samples purified from healthy tissues?

We thank the author for these questions and comments. We agree, the term ‘identified in breast cancer patient” is vague and as this rather relates to the source protein we have removed this information from the table as not fully serving the purpose.

To address the question on tumor antigen classification we searched the ‘CTDatabase’ for potential cancer testis antigens in our list of peptides. We did not identify any (orthologs of) human CT antigens. Thus, the identified peptides rather originate from overexpressed source proteins. However, we do not have experimental validation to support the overexpression of the source proteins in our model. We initially chose to use the term ‘Tumor associates antigens’ for our MHC-I ligands to make a clear distinction that we are not describing neoantigens in our study. The main text in refined accordingly (page 8, line 335).

In order to obtain more detailed information for the peptides presentation in normal tissues we have now searched publicly available database for the information on peptide HLA-presentation in various normal tissues. We utilized the ‘HLA Ligand Atlas’ as the resource of naturally presented HLA ligands in a comprehensive collection of human normal tissues. This resource contains information on HLA-presented peptides in 29 normal tissue types.

Out of the 21 peptides tested four peptides (FYITSRTQF, KYLATLETL, KYQAVTATL, NYQDTIGRL) were identified also in human normal tissue ligandomes (see figures below for details).

In retrospect, we agree this type of analysis would have been important to carry out prior to further experimentation and the information on peptide human normal tissue presentation is now included in the modified Table 1.

Of note, the ERV-peptides were not found to be HLA-presented in any normal tissue data deposited to the HLA Ligand Atlas- resource.

TAAs in group 4 were selected based on similarity to viral epitopes. This should be clarified in the text and in the table. Were any of these peptides (or their similar viral epitopes) shown to be immunogenic in other studies?

We have added the clarification for the peptide selection based on similarity to viral sequences in the main text (page 8, line 338). To address the question whether any of the peptides were shown to be immunogenic in other studies we searched the peptides through the IEDD database (Immune Epitope Database and analysis resource).

Each peptide within the TAA group 4 is described within the IEDB database and identified as ‘epitope with positive cellular MHC/mass spectrometry ligand presentation’. However, no other assays were reported for the peptides. In other words, each of these peptides has previously been identified as MHC-I ligand in a separate study using experimental approach similar to our study, but none of the peptides in TAA group 4, to our knowledge, has been tested experimentally for immunogenicity in these other studies.

Figure 4: which TAAs were used in these experiments?

The used TAA peptide sequences and their respective source proteins are provided in the materials and methods (4 page, line 200).

The authors should explain the unexpected results where ERV-PeptiCRAd with anti-PD1 did not induce antigen-specific immune responses and the ERV-PeptiCRAd by itself did (Figure 5b), while immunization with the TAA-peptiCRAd induced resposes only when anti PD1 was included (Figure 4b).

We thank the reviewer for a relevant question. Indeed, oncolytic vaccine therapy responses are improved when combined with immune checkpoint inhibitors (ICI). Likewise, we have previously shown that PeptiCRAd-mediated tumor control is improved by ICI combination (Ylösmäki et al., DOI:https://doi.org/10.1016/j.omto.2021.02.006 and Feola et al., doi: 10.1080/2162402X.2018.1457596).

We believe one explanation to the apparent discrepancy in our results is partly biological partly technical. The PeptiCRAd vaccine platform is relatively novel and as such immunologically not yet fully characterized. However, we anticipated it is possible we do not obtain very robust differences in the immunological responses between PeptiCRAd and PeptiCRAD + a-PD-1 group based on our previous experience. In our experience in the 4T1 mouse model the antigen-specific immunological responses are in general much weaker for the tumor antigens in comparison to the immunodominant antigen (gp70/AH-1). Furthermore, we were able to include only three randomly chosen mice from each treatment group for the immunogenicity test. It may appear that as individual mouse responses are variable, and the sensitivity of the present experiment is not optimal, even small changes may appear pronounced in the final result. Additionally, with the current experimentation we cannot rule out the possibility of antigen or epitope spreading playing a role in the treatment responses. Clearly, thorough experimentation would be required to understand this apparent discrepancy in more detail.

Line 103: SepPac should be Sep-Pak.

This is now corrected (page 3, line 111).

Line 105: please confirm that 0.11% FTA was used for the wash.

We apologize for the mistake, the misspelling is corrected and is now ‘0,1% TFA’ (page 3, line 112).

In figure 1b, the length distribution of the eluted peptides suggests a relatively high proportion of contaminating peptides (12-14 mers), this could be mentioned in the text.

Indeed, it is generally thought that the MHC-I -restricted epitopes are of a defined length of 8-11mers depending on the MHC-I allotype. However, some examples of unconventional, longer peptides capable of MHC-I presentation is presented in the literature.

We explored the predicted binding affinity of 12mer towards the relevant MHC-I within our dataset. We found that only 10 of the 12mers bind MHC-I with a predicted affinity strong enough to classify them as potential H-2-Kd ligands. On the other hand, these bioinformatic tools are not optimal for the longer peptides. However, as the MHC-I peptides are typically accepted to be 9 to 11 amino acid long we also believe the longer peptides may well represent contaminating peptides. This point is now included in the text on page 5, line 257.

Round 2

Reviewer 1 Report

The study is very informative and important. However, it still has some issues that need to be fixed before publishing. All these issues are minor, but a large number of them reduce the quality of the manuscript and downgrade excellent experimental work. My comments and revision suggestions are below.

Results

  • Supplementary Figure 1 presents a very good outline of all the work that has been done in the study. I think it deserves to be promoted to Figure 1 instead of being in supplementary material.
  • Regardless of long description of peptide selection approaches in the section entitled 3.1 entitled “Direct identification of tumor associated antigens in mouse triple-negative breast tumor” it is still unclear how the actual peptide filtering was done. How authors of the study manage to select a few peptides from so many? It still very difficult to understand how the peptides for pools one, two and three were selected from a few hundreds of identified candidates. There is a good explanation for pool 4, however. “one group (peptide pool #4) was selected based on in-house developed tool to score peptide immunogenicity using peptide high similarity to viral epitopes.” What about other pools, peptides from which are presented in Table 1? Perhaps, for clarity, a graphical scheme with selection steps for filtering out less promising peptide candidates will be useful.

  • Figure 5. “ERV-PeptiCRAd + a-PD-1” is almost invisible on the graph.
  • Line 328. The relevant sentence is unclear. The newly introduced word “first” makes it difficult for understanding.

Material and methods

The lines 191-194 (page 4) contain poorly formulated and unclear sentence: “These respond to the initially discovered FYLETQQQI, SYHPALNAI and KYQAVTATL originating from Brap, Birc6 and Rpl13a, respectively for the Figure 4 experiment and FYLPTIRAV and TYVAGDTQV originating from the ERV peptide in Figure 5.

  • It is not clear to what “These respond” refers.
  • This sentence lacks the word “peptides” as a category for the list of FYLETQQQI…It also lacks the word “genes” as a category for the list of names such as Brap, Birc6 and Rpl13a.
  • In addition, instead of “Figure 4 experiment” I suggest the "experiment shown in Figure 4".

On line 187, I suggest room temperature instead of RT. There are too many abbreviations in the text, which slow down reading and understanding.

Duplications and images without titles

There are some images and tables that are presented in duplicates. Among them are Figure 1 and Table 1. There are also some images without titles and legends on page 21 and page 25.

Reviewer 2 Report

see attached file
